# Healthcare Professionals’ Perceptions and Experiences of Ageism: A Qualitative Study

**DOI:** 10.3390/ijerph22030350

**Published:** 2025-02-27

**Authors:** Jiyoun Kim

**Affiliations:** College of Nursing, Kyundong University, 815 Gyeonhwon-ro, Munmak, Wonju 24695, Gangwondo, Republic of Korea; iscraa08@kduniv.ac.kr; Tel.: +82-033-738-1401; Fax: +82-033-738-1457

**Keywords:** ageism, geriatric healthcare professionals, grounded theory, aging discrimination, respect for elderly

## Abstract

This study focused on the experiences and perceptions of geriatric healthcare professionals. The research aimed to identify ageism, examine the influencing factors, explore the desired attitudes of geriatric healthcare professionals, and identify institutional changes required for age-integrated efforts and strategies to eliminate these barriers. Interviews with two physicians and eight nurses were conducted in South Korea from August to November 2023, with each session lasting about 1 h, and the transcripts were analyzed using the grounded theory by Strauss and Corbin. The analysis yielded 11 categories, 20 subcategories, and 120 concepts. The central phenomenon was “Ageism that is conducted implicitly and covertly”. Causal conditions affecting the development of ageism were “Difficulties related to the perceived characteristics of older adults” and “Extra burden for older patients and their families”. Contextual conditions were “Provider’s aging anxiety” and “Personal experience about older patients”, and interventional conditions were “Insufficient regular education aimed at ageism prevention” and “Insufficient staffing and resources”. The action/interaction strategies were “Not perceived as a critical issue” and “Perception that it is difficult to change”. After a thorough analysis and materializing of these concepts, the following prevention measures were proposed: “Need for regular education regarding the care of older patients (including ageism) in the nursing educational curriculum” and the “Need for sufficient staffing and resources”. According to the results of this study, medical professionals must satisfy the healthcare needs of the elderly by understanding the proper aging process and tailoring their approach to the specific characteristics of older individuals. To achieve this, organizations should provide adequate resources and personnel.

## 1. Introduction

The two fundamental pillars of healthcare coverage for older adults in South Korea are the National Health Insurance (the oldest social insurance program) and medical benefits for individuals with low incomes. In South Korea, there is no healthcare program or system specifically dedicated to older patients, such as “Medicare” in the United States or “Medical Care System for the Elderly in the Latter Stage of Life” in Japan. However, Korean health insurance coverage for older individuals has recently been extended to include certain medical services such as coverage for denture health insurance and an outpatient flat fee system for older patients [1]. Due to the natural cycle of life, older adults aged 65 years and over account for a substantial portion of healthcare utilization, with 43.4% of all healthcare spending in 2021 [2]. Given the rapid aging of Korean society, there is an increasing need for equitable health coverage for older adults, who are becoming an increasingly significant demographic group within society.

According to Butler [3], ageism is defined as a type of social and interpersonal discrimination imbued with stereotypes based on advanced age. Unlike racism and sexism, the attributes of which undergo minimal change over a lifetime, ageism differs in that age changes over time, rendering anyone susceptible to discrimination, likewise prone to discrimination. In ageism, negative beliefs and attitudes about the attributes of older individuals form the foundation for discriminatory behaviors. While not always detrimental, such beliefs and attitudes can lead to undesirable consequences when applied solely based on age, disregarding unique characteristics [4].

Naturally, each elderly person experiences different aging rates and health status; however in general, not only in the treatment of diseases, but also during the normal course of aging, older individuals experience physical, psychological, and social degradation and have an essential “encounter” with healthcare professionals in addressing these issues and receiving assistance. In this process, healthcare professionals significantly influence various aspects, including physical health, emotional functioning, and life expectancy, through their roles. Specifically, older adults in the final stages of life experience physical and psychosocial decline due to high morbidity and accelerated aging processes, leading to limitations in daily activities, which subsequently causes them to become “dependent” and require assistance from others [5]. However, healthcare professionals’ generalized attitudes towards older adults can result in neglecting their individual preferences and specific health requirements, despite the heterogeneity within the elderly population. This standardized treatment approach has been linked to negative health outcomes among elderly populations, underscoring the need for a more personalized and nuanced approach in geriatric care. Several studies have reported that geriatric patients experience discrimination based on age in healthcare, either explicitly or implicitly, compared to younger individuals, which contributes to negative health outcomes in older populations [6,7].

In South Korea, it has been postulated that healthcare professionals lack bonding with older individuals due to the nuclear family system that has evolved concurrently with the rapid aging of the population. These professionals possess limited holistic education encompassing psychological, family, and socioeconomic aspects of geriatric patients in addition to their physical characteristics, which contributes to the prevalence of ageism [8,9]. However, research indicates that some healthcare professionals reported being unfamiliar with the term ageism or perceived themselves as non-discriminatory [10].

Although there have been papers [10,11] analyzing interviews on ageism for nurses and nursing students, there is a lack of discussion on, not only the level of ageism, but also policy alternatives to reduce it, from psychological issues at the individual level, such as aging anxiety, for Korean doctors and nurses to questions about organizational issues in their medical institutions.

This study focused on the experiences and perceptions of geriatric healthcare professionals who treat older patients. The study aimed to identify if ageism was present in this process, examine the influencing factors, and explore the desired attitudes of geriatric healthcare professionals, as well as the institutional changes that are required for age-integrated efforts and strategies to eliminate these practices.

## 2. Materials and Methods

This study employed qualitative research methods to investigate the experiences of physicians and nurses in treating and caring for geriatric patients and exploring the influencing factors. This approach was selected because it enables the researcher to capture authentic accounts by directly interviewing healthcare professionals. Second, moral judgments about ageism are minimized, allowing the researcher to gain insight into the perceptions of healthcare professionals on ageism and their subjective experiences of its existence.

Among qualitative research methods, the grounded theory approach was selected. This method attempts to develop the theory of a particular phenomenon inductively through a series of systematic processes based on symbolic interactionism [12]. The grounded theory method enables the researcher to gain an in-depth understanding of how healthcare professionals comprehend, respond to, and act upon the phenomenon of ageism against older individuals. Additionally, it facilitates the discussion of policy measures to prevent ageism by considering the processes and contexts leading to this phenomenon.

The study presents a final model derived from data collected between August 2023 and November 2023. The study used purposeful sampling, with the initial participant recommended by my previous colleagues, and then nine other participants were also introduced to the researcher in a snowball sampling method by the previous participants. During a phone call, the researcher introduced himself, explained the aims of the study to them, and arranged an appointment with them if they expressed a verbal willingness to participate in this research. In the first meeting, written informed consent was obtained from the participants. Sampling continued until a theoretical saturation was achieved after the first interviews, with additional telephone interviews conducted for participants whose responses were unclear in content or meaning. Each participant’s initial interview lasted approximately one hour and was conducted at their institutional offices in South Korea.

The coding process based on grounded theory consists of open coding, axial coding, and selective coding. This process enables systematic data analysis and the inductive development of theory [12]. Based on grounded theory, open coding was conducted to closely examine the data, identify, label, and categorize phenomena. During this process, the categories and subcategories discovered were related according to the paradigm.

The inclusion criteria for participants were as follows: (1) a physician or nurse and (2) over 3 years of work experience at hospitals with more than 50% geriatric patients (over 65). Of the physicians and nurses who met the inclusion criteria, those who agreed to participate in the study were interviewed at randomly selected hospitals. The interviewees were two physicians and eight nurses, aged between 30 and 49 years, comprising four females and six males. Data were collected through in-depth, semi-structured, and face-to-face interviews with open-ended questions, observations, and field notes. The main interview questions were (1) personal experiences with older patients, (2) thoughts on aging and death, (3) feelings and thoughts regarding older individuals, (4) attitudes toward geriatric patients in the clinical setting, (5) differences from patients of other age groups, (6) thoughts on ageism, and (7) problems related to ageism in the clinical setting and ways to improve them.

In the present study, four supporting processes of trustworthiness were applied, namely, credibility, dependability, conformability, and transferability [13]. To enhance credibility, the research findings and interpretations were reviewed by peer researchers with extensive experience in qualitative research. Dependability was ensured by having the researcher re-code the same data after a certain period to verify consistency. Additionally, confirmability was strengthened by comparing the research findings with other research contexts to assess similarities, which were then documented. Approval for the study was obtained from the ethics committee of the nursing school at Kyungdong University (the no. IRB 2022-11, 19 September 2022). Written informed consent was obtained from all participants. Two female researchers assisted in verifying the codes extracted from the script in the study, although only the interviewer had contact with the participants; therefore, only the interviewer was aware of the identity of the participants throughout the study.

## 3. Results

The central phenomenon was “Ageism that is conducted implicitly and covertly”. Causal conditions that are events that allow ageism to manifest or develop were “Difficulties related to the perceived characteristics of older adults” and “Extra burden for the geriatric patients and their families” perceived by medical personnel; contextual conditions included in a specific list of properties in which a phenomenon lies were “Provider’s aging anxiety” and “Personal experience regarding older patients”; and interventional conditions that act to promote or suppress an action or interaction strategy taken within a specific context were “Insufficient regular education for ageism prevention” and “Insufficient staffing and resources”, while action/interaction strategies that have continuous and process characteristics were “Not perceived as a critical issue” and “Perception that it is difficult to change”.

The following Table 1 is an analysis based on the paradigm model of grounded theory after coding the data.

### 3.1. Central Category: Ageism That Is Carried out Implicitly and Covertly

A central phenomenon represents ‘what is going on here’ and is a central thought or event controlled by action/interactional strategies [12]. In other words, it refers to the central thoughts of healthcare professionals that are formed by causal and contextual conditions, and the central category identified in this study was ‘Ageism that is carried out implicitly and covertly’.

Some participants had not previously encountered the term ageism, and others had heard the term but requested clarification or examples. While most participants acknowledged that discrimination against older adults exists in society and the workplace, they denied or expressed disapproval of the occurrence of discriminatory behavior against geriatric patients by healthcare professionals such as doctors and nurses in the clinical setting. Paradoxically, most expressed difficulties in treating and caring for geriatric patients in clinical settings, highlighting negative characteristics of geriatric patients such as being impatient, stubborn, and uncommunicative, and demonstrating a clear preference for younger patients. The study [10] also reported that healthcare professionals agreed that certain characteristics of older patients impacted their work and service delivery, indicating that they possessed unintentional discriminatory perceptions. Discrimination against older adults was reported in the following forms.

#### 3.1.1. Communication with No Respect (Treating an Older Individual as a Child)

Healthcare providers responded that they often avoided questions or requests from older patients by moderately ignoring them. They assumed that the geriatric patient would have difficulty comprehending despite detailed explanations or that most of the questions would be unnecessary. Healthcare providers also appeared to believe that it was best to ignore geriatric patients because they were stubborn, such as asking the healthcare professionals to treat themselves first, exaggerating their pain, being demanding, or engaging in lengthy discussions about irrelevant matters. Sometimes, the providers would get angry or annoyed, although most often, they would feign not hearing them.

“But when that happens, even when it really is not, it makes me angry, so I say it louder. “Please be patient a little. Please wait’ (Nurse_D)”.

“I think it is just because they are older and a little bit cognitively impaired, and they cannot communicate with each other in a reasonably good way, and then they are just like saying, uh…so…. Consequently, physicians may inadvertently adopt a less considerate approach when addressing these patients, potentially due to a subconscious assumption that such patients believe that this is somebody who is not going to hurt me, even if they treat the patients this way. This may sometimes occur (Doctor_B)”.

They also reported that they performed treatments or nursing care without sex sensitivity for geriatric patients or those they assumed were cognitively impaired.

“When I go to the ICU, I do not know who they are because they are just lying there, so I just do it on my own. One time, I (a male nurse) was performing a routine procedure (foley insertion) and encountered an unexpected reaction from an older female patient who, upon seeing me, screamed and expressed significant distress, saying, ‘Oh, my God’, which prompted me to withdraw. In retrospect, I realized that while I had approached the task with professional detachment, the patient’s perspective of my sex had not been considered (Nurse_E)”.

#### 3.1.2. Double Discrimination Due to Complex Factors, Including Age

The study revealed that participants’ discriminatory behaviors were not solely influenced by the patient’s age but also by a combination of peripheral factors, including the socioeconomic level of the older patient (i.e., their appearance and education) and whether the patient was accompanied by an influential caregiver. However, these discriminatory behaviors manifested in relatively diverse forms, depending on the individual’s experiences, work environment, education, and other factors.

Double discrimination based on the combination of age and economic status creates practical discrimination in access to care, with economically disadvantaged older patients facing significant limitations regarding the choice of medications, wards, and service levels compared to their more affluent counterparts. It is not uncommon for them to be denied treatment, either by themselves or by their guardians, with financial responsibility, particularly if they incur out-of-pocket expenses.

“However, exceptions exist. Financial resources are a determining factor. One of the reasons why our hospital is so polarized is that our VIP ward is expensive. Superior amenities are provided, and the quality of care and nursing differ considerably. (Nurse_G)” 

“Many individuals who are eligible for healthcare receive government subsidies and are unwilling to incur additional expenses for non-covered treatments. Despite the existence of more effective methods to control pain, these individuals may be willing to endure discomfort and decline non-covered treatments to conserve financial resources. (Nurse_B)”

“When patients present with severe conditions such as pneumonia or sepsis, even in the presence of a guardian, they may be sensitive and refuse to take additional tests or receive fluids based on financial considerations. It appears that patients should have access to personal funds. (Nurse_D)”.

#### 3.1.3. Suboptimal Care for Older Patients: Over-Treatment or Under-Treatment in Medical Care

Medical professionals reported difficulties in determining the necessity for tests and treatments, as well as the appropriate level of aggressive treatment for geriatric patients. The level of treatment may sometimes be influenced by the individual physician’s perspective on aging or the hospital’s interests, potentially leading to suboptimal care for older patients. Regular hospitals operating under the fee-for-service system have incentives for over-treatment, whereas nursing hospitals operating under the comprehensive medical fee system have incentives for under-treatment in medical care. Geriatric patients may be unable to fully trust their physician’s medical decisions or actively express their preferences, thereby tending to be over-treated or under-treated due to the aforementioned incentives for healthcare institutions.

“There are numerous factors to consider as a physician, including the patient’s economic level, expectations, the benefits and risks of the treatment, and the prognosis. In particular, I should consider the age and the specific aspects of the patient’s surgery. Consequently, every patient’s situation lies somewhere between over-treatment and under-treatment in medical care. I believe that physicians frequently encounter these situations (Doctor_A)”.

“It is challenging to ensure that older patients undergo the necessary tests they need or unnecessary ones. It’s often due to financial constraints and limited understanding (Doctor_B)”.

#### 3.1.4. Low Respect for Older Individuals’ Self-Determination

For geriatric patients, important medical decisions are often made in discussion with the primary caregiver, who is also the financial caretaker. The eldest of the children or the financial caretaker is often the primary decision-maker. Decisions about the length of hospital stay, surgeries, and treatments are mostly made between the primary caretaker and the attending physician.

“It appeared that the older patient had limited autonomy in decisions regarding end-of-life care or treatment options. (omitted) His children seemed to be the primary decision-makers (Nurse_B)”.

“The patient says, ‘I don’t want a nursing home’, and the child says, ‘I can’t take care of my parents, so you should stay in the hospital’. This is the most common case (Nurse_G)”.

“The decision is always made with the guardians because they’re paying for it (Nurse_E)”.

Even when doctors need to convince patients of a course of treatment or method, they often do not take the time to persuade them; instead, they create an atmosphere of coercion and insist on the decisions that the doctors make.

“Sometimes we can’t keep persuading them. We just have to force them to do it. I just have to force them. I just need to say, ‘You must do this’. The elderly patients have many experiences in meeting with doctors who care for patients in an old style. If I can’t persuade them any longer, I can’t help forcing them to follow. I say, ‘I have a reason to say this, so you must follow’ If I can’t give up on persuasion, I just go with an emotional appeal (Doctor_B)”.

### 3.2. Overcoming Ageism

#### 3.2.1. Need for Regular Education About the Care for Older Patients (Including Ageism) and the Nursing Educational Curriculum

The curriculum for healthcare professionals and education regarding the specific needs of geriatric patients focus primarily on physical changes, with limited emphasis on ageism. However, there is general consensus regarding the necessity for training on the physical, mental, and cultural specificities of geriatric patients encountered in the treatment and care in clinical practice, as well as on the appropriate ethical treatment of these patients. Furthermore, the majority of participants demonstrated a lack of awareness with respect to ageism in clinical practice. Some participants even reported unfamiliarity with the term “ageism”.

“The geriatric nursing curriculum was too rudimentary. We primarily focused on physical characteristics, sexual function, and similar topics. I think we learned mostly about physical changes in older adults, such as alterations in sleep patterns (Nurse_A)”. “Education is essential; however, it is important to acknowledge our own shortcomings. We require education to enhance our empathy towards older individuals. I believe we need to focus on education, not only among us but with the older patients elderly (Nurse_G)”.

“I believe the implementation of ageism education would facilitate smoother interactions (Nurse_H)”.

#### 3.2.2. Need for Sufficient Staffing and Resources

Nurses reported that geriatric patients exhibited greater needs than patients of other age groups and required more time and effort for communication and education. With inadequate staffing and schedule constraints, nurses felt burdened to provide appropriate care for geriatric patients.

“I believe that with sufficient time allocated to patient care, I would be able to perform more effectively. Typically, when pressed for time, I attempt to expedite tasks (omitted). It is crucial to have adequate time for each patient’s care, and staffing issues are also a concern (Nurse_B)”.

“Older adults tend to move more slowly, which necessitates additional time to complete tasks. Consequently, this reduces the time available to provide assistance. Staffing issues are also a contributing factor (Nurse_D)”.

Healthcare professionals perceived that geriatric patients were less patient and tended to exaggerate their pain when the treatment process was not explained in detail or when they reported experiencing pain.

However, most participants in the study did not believe that they were discriminating based on age, explaining that the age of the patient did not overtly drive these discriminatory behaviors. In response to frequent questions from geriatric patients, healthcare providers reported that they often responded by saying that they were tailoring their care and treatment to the characteristics of each geriatric patient. By answering that way, they tended to evade questions or requests. Healthcare providers often considered that geriatric patients would have difficulty understanding even though the providers gave detailed explanations. Otherwise, they thought that the questions of older patients were not significant. Sometimes, the geriatric patients were so insistent that they wanted to be attended to first, complained of exaggerated pain, were demanding, or talked at length about irrelevant matters. As a result, it appeared that healthcare professionals felt it was best to ignore them. While they sometimes became angry or irritated, they mostly pretended not to hear them.

Conversely, some healthcare professionals attempted to approach older patients in an amiable manner based on their individual personalities. Older adults tend to value relationship building, particularly with medical staff at their most frequently utilized healthcare facilities. Addressing older patients by their names or engaging in conversations about their personal conditions or circumstances related to their illness can facilitate the maintenance of a close relationship.

## 4. Discussion

The primary category identified in this study was ageism, covertly practiced by healthcare professionals against geriatric patients and is identified as “Communication with no respect” and “Over-treatment or under-treatment during medical and nursing care”. This phenomenon is not solely driven by age but rather by a combination of factors such as the patient’s appearance, education, and economic status.

A major cause of this ageism includes the stereotypical behaviors of older individuals as perceived by healthcare providers during treatment. For example, geriatric patients often exhibit less patience, are more stubborn, and present with more complex and severe illnesses due to the aging process and chronic conditions, which were reported to render treatment and care more challenging than in other age groups.

For the contextual condition, healthcare providers’ personal experience with older patients influences ageism, with numerous positive contacts, whether personal or professional, offsetting ageism. As a contextual condition, aging anxiety as perceived by healthcare professionals was identified, with younger generations viewing adults differently from themselves and subtly not identifying with them as equivalent human beings. Aronson’s research [14] suggests that old age is the fate of all humans, although it is not a fate they anticipate. In some ways, death may be perceived as more appealing due to the potential deterioration of quality of life with aging, and because the gradual progression towards death, rather than an abrupt occurrence, elicits greater apprehension. This aging anxiety can cause healthcare professionals to have negative emotions when faced with geriatric patients. Thus, aging anxiety affects younger people’s attitudes and behaviors toward the elderly, creating additional anxiety about aging itself [15].

To provide quality care for geriatric patients, it is important to implement preparatory measures. Healthcare providers report that the burden of care for geriatric patients is 1.5 to 2 times greater than for patients of other age groups due to the complexity and severity of geriatric patients’ illnesses. While caring for geriatric patients, work stress from the endless demands of daily care and the persistent anxiety and tension regarding safety incidents are the primary contributing factors to burnout among nurses [16]. Medical fees in Korean nursing hospitals are based on the ‘per diem’ system, under which hospitals receive a flat medical fee for each geriatric patient. Consequently, hospitals lack incentives to provide high-quality services. Moreover, severely ill geriatric patients who place a heavy burden on healthcare services are even more marginalized from quality care. Since the ‘fee-for-service’ system compensates regular hospitals, they tend to provide over-treatment in medical care, including expensive non-covered items and unnecessary diagnoses and treatments. Therefore, there is a high demand for the realistic application of medical fees in the treatment and care of geriatric patients [9].

Participants also emphasized the need for regular education and training for healthcare providers, which should encompass training on recognizing and addressing ageism, in addition to medical knowledge related to physical and mental changes with aging [11]. Other studies, consistent with the present study, reported that patients with significant cognitive decline or geriatric patients are often insensitive to care or treatment that may be sexually offensive [17,18]. To prevent this, it is necessary to increase cognitive sensitivity regarding sex among geriatric patients.

Moreover, some participants in this study reported that their personal experiences with older family members or their positive and negative experiences with geriatric patients influenced their subsequent care and treatment practices. As a result, nurses’ positive attitudes toward older patients may improve the overall practice of geriatric care. Consequently, it is necessary to develop strategies to increase healthcare providers’ positive attitudes toward older individuals. Some studies revealed that nurses with greater knowledge about older patients had more positive attitudes [19], and others revealed that nurses’ attitudes toward older adults were different depending on their experiences of living with the elderly [20]. These studies suggest that, not only geriatric nursing education, but also contact between nurses and older adults can provide opportunities to cultivate positive attitudes toward older adults [21,22].

This study is significant as it provides foundational data for future policy discussions aimed at reducing ageism in healthcare practice. However, to develop more specific and actionable policy recommendations, it is important to acknowledge the imbalance in the sample composition, with a markedly smaller number of physicians compared to nurses. Future research should consider either comparing the two groups while accounting for the distinct characteristics of each professional role, or conducting separate analyses for each group to ensure a more nuanced understanding.

## 5. Conclusions

Based on this study’s results, the following suggestions are proposed. First, healthcare providers need to acknowledge older persons as individuals with diverse health needs and characteristics rather than as a homogeneous group possessing typical characteristics. Second, to ensure that the previous proposal is implemented effectively, strategies to address ageism in healthcare settings should include the establishment of hospital environments, staffing, and training that consider specialized treatment for geriatric patients. Additionally, nursing practices should be customized to meet the specific needs of older patients.

## Figures and Tables

**Table 1 ijerph-22-00350-t001:** Formation of ageism in clinical settings.

Subcategories	Categories	Paradigm Element
Communication difficulties (cognition/physical) Ambivalence toward older individuals (compassion, sympathy/displeasing, frustration, among others)	Difficulties related to the perceived characteristics of older adults	Causal condition
Complicated and longer medical/nursing process Hard job demanding a lot of touches/explanations	Extra burden for the geriatric patients and their families	
Communication with no respect (treating the elderly as a child) Double discrimination due to complex factors including age Over-treatment or under-treatment in medical care Low respect for self-determination by older individuals	Ageism that is conducted implicitly and covertly	Central phenomenon
Aging anxiety (aging is worse than dying) Degree of prejudice toward older individuals	Personal experience and perception about older individuals	Contextual condition
Personal bad experiences with elderly		
No regular education about elderly care (including ageism)	Insufficient regular education for ageism prevention	Intervening condition
Not included in the nursing educational curriculum Insufficient nursing time Weighted systemic support for the care of older	Insufficient staffing and resources	
Not recognizing it as a problem No awareness that it is an issue	Not perceived as a critical issue	Action/interactional strategies
Recognizing it as an issue, although it cannot be addressed Regrettable but challenging to rectify	Perception that it is difficult to change	
Unmet physical care needs Low-priority patients Informational inequality The perception that one is incapable of making decisions for oneself	Delayed recovery Interactional injustice Loss of autonomy	Consequences
Unfair care (delayed health care/social discrimination)	Procedural injustice	

## Data Availability

The qualitative dataset/transcriptions of narratives is not publicly available due to ethical restrictions and privacy issues.

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
