# Peer review of "Healthcare Professionals’ Perceptions and Experiences of Ageism: A Qualitative Study"

_ijerph, 2025, doi:10.3390/ijerph22030350_

Round 1
Reviewer 1 Report
Comments and Suggestions for Authors
The study reported in the paper relates to the important issue of ageism in the provision of geriatric care. The study was conducted using grounded theory and interviews with 11 healthcare professionals. Some interesting findings emerged but I hope the paper can be strengthened by responding to the comments below.
Introduction:
The introduction would be strengthened by including discussion about the heterogeneity of the older population. The third paragraph starts by describing a standardised trajectory of ageing – not all people will follow the same ageing patterns.
The gap in the current literature that the study sets out should be explicitly articulated.
In the last paragraph, I would suggest rephrasing the line “The study aimed to identify the presence of ageism in this process” to something more like “…aimed to identify if ageism was present in this process…”. Although prior research suggest that there is ageism in healthcare, it should not assumed to be present in all cases.
Methods:
More information is needed to explain how the grounded theory process was used. How were the casual, contextual and institutional conditions materialised?
More information about the data collection is needed. How were the participants recruited? How was the study promoted? How many different sites did they work across? Why were only 11 interviews conducted? How long were the interviews?
Results:
Maybe a table would be useful to help show how the casual, contextual and institutional conditions were decided upon. These conditions are briefly mentioned in the Discussion but should also be reported in the results.
How are the conditions defined? For example, what is meant by the “Difficulties arising from characteristics associated with aging”? This sounds like a very general and potentially ageist statement in and of itself. Should it be “perceived characteristics”?
I would like to see more of the process that led to the central phenomenon.
In lines 120 and 238 there is a typo, the prevention measure should be “… in the nursing educational…” not “…and the…” (This is correct in the abstract.)
Is there a reason this recommendation is just for nurses as opposed to all health care professionals? As well as for older people/the general public as suggested by one of your participants.
The last two paragraphs of the results do not seem to relate to “sufficient staffing and resources”, do these findings around the ageist behaviours of health professionals relate more to the first recommendation relating to education?
Discussion:
Should low-respect and self-determination also be included in the first paragraph?
Again, should line 297 be “…geriatric patients are perceived to exhibit…” as opposed to often exhibit? I would avoid these generalisations.
Author Response
Comments 1: The introduction would be strengthened by including discussion about the heterogeneity of the older population. The third paragraph starts by describing a standardised trajectory of ageing – not all people will follow the same ageing patterns.
Response 1: I added
“Naturally, each elderly person experiences different aging rates and health status, however in general,” not only in the treatment of diseases but also during the normal course of aging, ” in 3rd paragraph
Comments 2: The gap in the current literature that the study sets out should be explicitly articulated.
Response 2: Agree. I added,
“Although there have been papers(Kim, J. et al. , 2016;Oh, H. I.et al., 2018) analyzing interviews on ageism for nurses and nursing students, there is a lack of discussion on not only the level of ageism but also policy alternatives to reduce it, from psychological issues at the individual level such as aging anxiety for Korean doctors and nurses to questions about organizational issues in their medical institutions” in 5th paragraph
Comments 3: In the last paragraph, I would suggest rephrasing the line “The study aimed to identify the presence of ageism in this process” to something more like “…aimed to identify if ageism was present in this process…”. Although prior research suggest that there is ageism in healthcare, it should not assumed to be present in all case
Response 3: Agree. I modified like what you suggested,
Comments 4: More information about the data collection is needed. How were the participants recruited? How was the study promoted? How many different sites did they work across? Why were only 11 interviews conducted? How long were the interviews?
Response 4: Agree. I added,
“The study presents a final model derived from data collected between August 2023 and November 2023. The study used purposeful sampling, with the initial participant recommended by my previous colleagues, then other 10 participants were also introduced to the researcher in a snowball sampling by the previous participants. During a phone call, the researcher introducedhimself explained the aims of the study to them, and arranged an appointment with them in case of their verbal desire to participate in this research. In the first meeting, written informed consent was obtained from the participants. Sampling continued until theoretical saturation was achieved after 25 interviews(each participant had 1~2 interviews and usually took about an hour), and no new conceptual codes were identified from the data”
Comments 5: Maybe a table would be useful to help show how the casual, contextual and institutional conditions were decided upon. These conditions are briefly mentioned in the Discussion but should also be reported in the results.
Response 5: Agree. I modified and added,
“The central phenomenon was “Ageism that is conducted implicitly and covertly.” Causal conditions that are events that allow ageism to manifest or develop were “Difficulties arising from characteristics associated with aging”and “Extra burden for the geriatric patients and their families” perceived by medical personne ; contextual conditions that is a specific list of properties in which a phenomenon lies were 'Provider's aging anxiety” and “Personal experience regarding older patients”; and interventional conditions that a condition that acts to promote or suppress an action or interaction strategy taken within a specific context were “Insufficient regular education for ageism prevention,” and ”Insufficient staffing and resources”; while action/interaction strategies that has continuous and process characteristics were “Not perceived as a critical issue” and “Perception that it is difficult to change.” After materializing them, the following ageism prevention measures were proposed based on them. The identified consequences that Consequence refers to the result of an action or interaction strategy were “Need for regular education about elderly care (including ageism) and the nursing educational curriculum” and “Need for sufficient staffing and resources.”
Comments 6: How are the conditions defined? For example, what is meant by the “Difficulties arising from characteristics associated with aging”? This sounds like a very general and potentially ageist statement in and of itself. Should it be “perceived characteristics”?
Response 6: Agree. I modified and added.
The central phenomenon was “Ageism that is conducted implicitly and covertly.” Causal conditions that are events that allow ageism to manifest or develop were “Difficulties arising from characteristics associated with aging”and “Extra burden for the geriatric patients and their families” “perceived by medical personnel”
Comments 7: I would like to see more of the process that led to the central phenomenon.
Response 7: According to my paper,entire 3-1 Paragraphs list detailed descriptions of the central phenomenon.3.1.1. Communication with no respect (treating an older individual as a child)/3.1.2. Low respect for older individuals’ self-determination/3.1.3. Suboptimal care for older patients: Over-treatment or under-treatment in medical care/3.1.4. Double discrimination due to complex factors, including age
Comments 8: In lines 120 and 238 there is a typo, the prevention measure should be “… in the nursing educational…” not “…and the…” (This is correct in the abstract.)
Response 8: agree. I changed ”prevention measures should be proposed”
Comments 9 : Is there a reason this recommendation is just for nurses as opposed to all health care professionals? As well as for older people/the general public as suggested by one of your participants
Response 9: agree. The author's suggestion is for medical professionals such as doctors and nurses. I clarified this in the conclusion.
Comments 10 :The last two paragraphs of the results do not seem to relate to “sufficient staffing and resources”, do these findings around the ageist behaviours of health professionals relate more to the first recommendation relating to education?
Response 10: agree. The placement of paragraphs was revised according to their importance and the point was made clearer as your advie. According to “the results of this study, the following some suggestions should be made. First,” healthcare providers need to acknowledge older persons as individuals with diverse health needs and characteristics rather than as a homogeneous group possessing typical characteristics. “Second, to ensure that the previous proposal is implemented effectively,” the strategies to address ageism in healthcare settings should include the establishment of hospital environments with facilities, staffing, and training that consider specialized treatment for geriatric patients, with nursing practices that are customized to the needs of older patients.

Reviewer 2 Report
Comments and Suggestions for Authors
Please, see the attachment.

Author Response
|
Comments 1: : by my opinion, the following aspects could be addressed.
1) To rewrite a little the sentences at lines 7-10, for greater clarity, e.g., (this is only an example) Thisstudy focused on the experiences and perceptions of ageism by geriatric healthcare professionals.The research aimed to identify related influencing factors, to explore the desired attitudes ofgeriatric healthcare professionals, and to identify institutional changes required for age-integratedefforts and strategies to eliminate these barriers --> “This study focused on the experiences and perceptions of ageism by geriatric healthcare professionals.The research aimed to identify related influencing factors, to explore the desired attitudes ofgeriatric healthcare professionals, and to identify institutional changes required for age-integratedefforts and strategies to eliminate these barriers” (in 1st paragraph) 2) To summarize the results without quotation marks (lines 13-24). --> Another reviewer said that the current form of the quota format is good, and I think I can see clearer results, so I decided to leave it as it is. I appreciate your understanding.
3) To add some lines regarding Conclusions (from par. 5). -->I added, “ According to the results of this study, Medical professionals must satisfy the healthcare needs of the elderly by understanding the proper aging process and tailoring their approach to the specific characteristics of older individuals. To achieve this, organizations should provide adequate resources and personnel.” in last part of abstract. 4) To add where and when the study was carried out. --> I added, “Comprehensive interviews were conducted a total of 25 times, with each session lasting approximately 1 hour with 2 physicians and 8 nurses, between August 2023 and November 2023”
|
||
|
Response 1: Thank you for pointing this out. I rewrited as what you suggested. |
||
|
Comments 2: Keywords: Authors could “respect” and “discrrmination”
|
||
|
Response 2: I agree. I added, Keywords:ageism; geriatric healthcare professionals; Grounded Theory; “aging discrrmination; respect for elderly”
Comments 3: Introduction: this section is short but well done. I would however suggest to begin with general concepts on ageism (lines 42-62), and to continue with some in focus on ageism and South Korea (line 29-40 and lines 63-70,to be integrated appropriately)
Response3: There may be slight inconsistencies in the flow of this section. However, the following paragraph provides a detailed explanation of the ageism observed among healthcare professionals in South Korea, which helps to maintain the overall coherence of the discussion. Your kind understanding in this matter would be greatly appreciated
Nest paragraph is In South Korea, it has been postulated that healthcare professionals lack bonding with older individuals due to the nuclear family system that has evolved concurrently with the rapid aging of the population. These professionals possess limited holistic education encompassing psychological, family, and socioeconomic aspects of geriatric patients in addition to their physical characteristics, which contributes to the prevalence of ageism (Han & Lee, 2010; Lee et al., 2014). However, research indicates that some healthcare professionals reported being unfamiliar with the term ageism or perceived themselves as non-discriminatory (Oh, Joo, & Kim, 2018).
Comments 4: Materials and Methods. This section by my opinion needs several adjustments, and should beintegrated/clarified with further (necessary) details. I would suggest the following. - To split the par. in 2 sub-paragraphs, e.g.: o 2.1. Study Design, Data Collection, and Ethics (current lines 77-109). Here Authors should: l add where and when the study was carried out; l also, they could clarify if they used a purposive sampling, i.e., where units/cases areselected due to particular characteristics allowing a deep exploration of thephenomenon of interest (Palinkas LA, Horwitz SM, Green CA, Wisdom JP, Duan N,Hoagwood K. Purposeful Sampling for Qualitative Data Collection and Analysis inMixed Method Implementation Research. Adm Policy Ment Health. 2015Sep;42(5):533-44. doi: 10.1007/s10488-013-0528-y) -->I added, “ The study presents a final model derived from data collected between August 2023 and November 2023. The study used purposeful sampling, with the initial participant recommended by my previous colleagues, then other 10 participants were also introduced to the researcher in a snowball sampling by the previous participants. During a phone call, the researcher introducedhimself explained the aims of the study to them, and arranged an appointment with them in case of their verbal desire to participate in this research. In the first meeting, written informed consent was obtained from the participants. Sampling continued until theoretical saturation was achieved after 25 interviews(each participant had 1~2 interviews and usually took about an hour in their office), and no new conceptual codes were identified from the data.”. l moreover, a definition of geriatric patients (line 78) should be provided (e.g. 65+years); --> I added, patients”(over 65)”
l inclusion criteria (lines 92-95): to add infos on selected hospitals, and type ofgeriatric patients (age, main disease to move infos at lines 95-96 (interviewees were two physicians and eight nurses,aged between 30 and 49 years, comprising four females and six males) in a Table 2to be put at the begin of Results as socio-demographic characteristics of the sample,also adding further aspects, if any (e.g., specialization) --> Thank you for your feedback. To clarify, the sample indeed consists of two physicians and eight nurses. However, Table 2 was removed after an additional submission. The participants are physicians and nurses working in long-term care hospitals without specialization)
l to explain how the processes of trustworthiness were assured, e.g.(these are examples): credibility of results by means of frequent peer de-briefingsessions among researchers with deep experience; transferability through a carefulpreliminary literature review; dependability and confirmability as accuratedescription of the study protocol and transparent/replicable procedures, in additionto collaborative discussion among researchers regarding particularly data analysis. --> I agree. I added, In the present study, four supporting processes of trustworthiness were applied, namely, credibility, dependability, conformability, and transferability (Yilmaz, 2013). “To enhance credibility, the research findings and interpretations were reviewed by peer researchers with extensive experience in qualitative research. Dependability was ensured by having the researcher re-code the same data after a certain period to verify consistency. Additionally, confirmability was strengthened by comparing the research findings with other research contexts to assess similarities, which were then documented.”
Comments 5: 2.2. Data Analysis (that is fully lacking). Here Authors should include the following (some examples): l method of qualitative analysis (e.g., the Framework Analysis Technique? as indepth reading of transcribed interviews identification of categories,coding/concepts, interpretation of the qualitative content); l thematic content analysis ; l manual qualitative analysis or using a software;presentation of categories in Table 1 (that could be moved from par. 3 Results to par.2. Materials and Methods); --> I added, “The coding process based on grounded theory consists of open coding, axial coding, and selective coding. This process enables systematic data analysis and the inductive development of theory(Strauss& Corbin, 1990). Based on grounded theory, open coding was conducted to closely examine the data, identify phenomena, label them, and categorize them. During this process, the categories and subcategories discovered were related according to the paradigm.” l to explain/anticipate here (and thus move from lines 124-126) the followingsentence: a central phenomenon represents 'what is going on here' and is a centralthought or event controlled by action/interactional strategies (Strauss & Corbin,1990). In other words, it refers to the central thoughts of healthcare professionalsthat are formed by causal and contecual condition”; l to add that the analysis of results was integrated by verbatim statements whichemerged in the transcription of the interviews;to add that each quotation (at least I suppose so) was coded by inserting the type ofrespondent (nurse or doctor) and the progressive intervie
--> I modified in the first paragrap of Result “The central phenomenon was “Ageism that is conducted implicitly and covertly.” Causal conditions that are events that allow ageism to manifest or develop were “Difficulties arising from characteristics associated with aging”and “Extra burden for the geriatric patients and their families” perceived by medical personne ; contextual conditions that is a specific list of properties in which a phenomenon lies were 'Provider's aging anxiety” and “Personal experience regarding older patients”; and interventional conditions that a condition that acts to promote or suppress an action or interaction strategy taken within a specific context were “Insufficient regular education for ageism prevention,” and ”Insufficient staffing and resources”; while action/interaction strategies that has continuous and process characteristics were “Not perceived as a critical issue” and “Perception that it is difficult to change.” After materializing them, the following ageism prevention measures were proposed based on them. The identified consequences that Consequence refers to the result of an action or interaction strategy were “Need for regular education about elderly care (including ageism) and the nursing educational curriculum” and “Need for sufficient staffing and resources.”
Comments 6: Results. This section is almost detailed for some aspects, but other information seems lacking, and someparagraph restructuring seems necessary, or at least useful for greater clarity. I would suggest the following. l Line 119: Authors mention “consequences”, but they indicate de factor “prevention measures”(this could be a typo). --> I agree. I rearranged “The identified consequences that Consequence refers to the result of an action or interaction strategy were “Need for regular education about elderly care (including ageism) and the nursing educational curriculum” and “Need for sufficient staffing and resources. “ After materializing them, the following ageism prevention measures were proposed based on them. ” in the last part of this paragraph. l Table 1: as already suggested above, this could be moved in the Methods section (topresent/anticipate the categories emerged from the analysis). Moreover, categories could bementioned in the order they are presented in the Results section, i.e., from central category, followedby causal conditions, contextual conditions, interventional conditions, action/interactional strategies,and consequences. Also, Authors should add prevention measures (that are missed in Table 1), i.e.,Need for regular education about elderly care (including ageism) and the nursing educational curriculum” and “ Need for sufficient staffing and resources . --> Thank you for your detailed and helpful feedback. Initially, we created a paradigm model and included a figure, but due to its overlap with other content and concerns about the paper's length, we ultimately decided to remove it with another reviewer. Additionally, the section stating, “Need for regular education about elderly care (including ageism) and the nursing educational curriculum” and “Need for sufficient staffing and resources”, has been newly composed to address the identified interventional conditions. While it may be insufficient, we kindly ask for your understanding as it is difficult to make major revisions to this part.
l Par. 3.1.: Authors could put concepts in sub-par. in the order they are presented in Table 1., i.e.: o 3.1.1. Communication with no respect (treating the elderly as a child) (this is already so) o 3.1.2. Double discrimination due to complex factors including age o 3.1.3. Over-treatment or under-treatment in medical care (this is already so) o 3.1.4. Low respect for self-determination by older individuals
--> As you mentioned, there was an error here. The order in the table and the order described in the main text were different. Therefore, the description has been revised to match the order in the table. 3.1.”2.” Double discrimination due to complex factors, including age 3.1.”4.” Low respect for older individuals’ self-determination
l Authors could add more sentences and quotations also on causal/contextual/interventional conditions, strategies, and consequences (aspects that are mentioned in Table 1). --> Yes, that is correct. Initially, the sentences and citations regarding causal/contextual/interventional conditions, strategies, and consequenceswere included in order in the original manuscript. However, for Part 3.2, we thought it would be better to structure the content based on policy alternatives, so we boldly removed those details. We kindly ask for your understanding regarding this decision.
Comment 7: Discussion: This section presents some critical aspects, since Authors discus some results which areindicated in Table 1, but which are only briefly listed in the text of the Results section (as already indicatedabove), without reporting some useful quotation (e.g. anxiety, burden of care, personal experience ofhealthcare professionals). Thus, I would recommend to include these results in the Results section (withrelevant quotations) in order to justify their discussion, and overall to discuss more the main results(Communication with no respect, Double discrimination, Over-treatment or under-treatment in medicalcare, Low respect for self-determination by older individuals, Need for regular education and sufficientstaffing and resources).
Response 7: I agree, so I modified and added.
“The central phenomenon was “Ageism that is conducted implicitly and covertly.” Causal conditions that are events that allow ageism to manifest or develop were “Difficulties arising from characteristics associated with aging”and “Extra burden for the geriatric patients and their families” perceived by medical personne ; contextual conditions that is a specific list of properties in which a phenomenon lies were 'Provider's aging anxiety” and “Personal experience regarding older patients”; and interventional conditions that a condition that acts to promote or suppress an action or interaction strategy taken within a specific context were “Insufficient regular education for ageism prevention,” and ”Insufficient staffing and resources”; while action/interaction strategies that has continuous and process characteristics were “Not perceived as a critical issue” and “Perception that it is difficult to change.” After materializing them, the following ageism prevention measures were proposed based on them. The identified consequences that Consequence refers to the result of an action or interaction strategy were “Need for regular education about elderly care (including ageism) and the nursing educational curriculum” and “Need for sufficient staffing and resources.”
Comments 8: Limitations: I would suggest to add at least a sentence in this regard at the end of the Discussion (e.g.., smallsample, and other aspects that Authors consider as such).
Response 8: I add this in the last part “This study is significant as it provides foundational data for future policy discussions aimed at reducing ageism in healthcare practice. However, to develop more specific and actionable policy recommendations, it is important to acknowledge the imbalance in the participant composition, with a markedly smaller number of physicians compared to nurses. Future research should consider either comparing the two groups while accounting for the distinct characteristics of each professional role or conducting separate analyses for each group to ensure a more nuanced understanding”
Comments 9: Data Availability Statement:Authors state that “No new data were created or analysed in this study. Data sharing is not applicable to this articlequalitative study that however produced qualitative results. Maybe it could be more appropriate, forinstance, to state that the data presented in the study are available on request from the corresponding author(if is this the case). Conversely, they can state that the qualitative dataset/transcriptions of narratives is notpublicly available due to ethical restrictions and privacy issues (again, if is this the case). Or other sentencescan be provided, corresponding to the real situation.
Response 9: I agree. I change. “Data Availability Statement: The qualitative dataset/transcriptions of narratives is not publicly available due to ethical restrictions and privacy issues.”
Comment 10: References Few of them are recent (only one on 2022). Moreover, references should be put in the rightformat (numbers in square brackets in the text, and in the style requested by the Journal in the final referencelist at the end of the paper).
Response 10: I made an effort to gather relevant references; however, I acknowledge the lack of recent literature in the references. I fully accept this constructive feedback and will address it accordingly.
I sincerely appreciate your meticulous review of my inadequate paper and the thoughtful advice you provided, complete with specific examples. Your feedback has been immensely helpful.
|
Round 2
Reviewer 1 Report
Comments and Suggestions for Authors
Thank you for responding to my comments, however my comments were not fully addressed and I think improvements could still be made.
Introduction:
Comment 1: I suggested that the introduction would be strengthened by including discussion about the heterogeneity of the older population. Half a sentence was added but not a discussion. I would dispute that there is a “normal course of aging” (Line 66).
Methods:
Line 119: This does not make grammatical sense: “and arranged an appointment with them in case of their verbal desire to participate in this research.”
Line 121: The following sentence does not make sense: “Sampling continued until theoretical saturation was achieved after 25 interviews(each participant 122
had 1~2 interviews and usually took about an hour in their office)”
How can 1 or 2 interviews with 11 participants equal 25? Why were participants interviewed multiple times? Were different questions asked? It is still unclear to me how and why the data was collected.
Line 134: “The interviewees were two physicians and eight nurses, aged between 30 and 49 years, comprising four females and six males.” This statement implies there were 10 participants, not 11?
Results:
Comments 6: How are the conditions defined? I still think that the condition of “Difficulties arising from characteristics associated with aging” needs further explanation.
In the abstract this condition is listed as “Difficulties arising from inherent characteristics of older adults” – there is a need for consistency. And I still suggest that this sounds like a very general and potentially ageist statement in and of itself. Should it be “perceived characteristics” or “associated characteristics”?
Comments on the Quality of English Language
The additions that were made need proofing.
Author Response
|
I sincerely appreciate your thorough review and valuable advice on my paper. Your feedback has significantly contributed to making it a better manuscript Response 1: Agree. I added accordingly “However, healthcare professionals' generalized attitudes towards older adults can result in neglecting their individual preferences and specific health requirements, despite the heterogeneity within the elderly population. This standardized treatment approach has been linked to negative health outcomes among elderly populations, underscoring the need for a more personalized and nuanced approach in geriatric care.” line 65 |
||
|
Comments 2: Line 119: This does not make grammatical sense: “and arranged an appointment with them in case of their verbal desire to participate in this research.” |
||
|
Response 2: Agree. I modified accordingly “arranged an appointment with them if they expressed a verbal willingness to participate in this research. ” line 113
Comment 3: Line 121: The following sentence does not make sense: “Sampling continued until theoretical saturation was achieved after 25 interviews(each participant 122 had 1~2 interviews and usually took about an hour in their office)” How can 1 or 2 interviews with 11 participants equal 25? Why were participants interviewed multiple times? Were different questions asked? It is still unclear to me how and why the data was collected
Response 3: I agree, so I modified “Sampling continued until theoretical saturation was achieved after the first interviews, with additional telephone interviews conducted for participants whose responses were unclear in content or meaning. Each participant’s initial interview lasted approximately one hour and was conducted in their office.”
Comment 4: Line 134: “The interviewees were two physicians and eight nurses, aged between 30 and 49 years, comprising four females and six males.” This statement implies there were 10 participants, not 11? ions asked? It is still unclear to me how and why the data was collected
Response 4: I agree, I put wrong number. I changed it.
Comment 5: Line 134: How are the conditions defined? I still think that the condition of “Difficulties arising from characteristics associated with aging” needs further explanation.
In the abstract this condition is listed as “Difficulties arising from inherent characteristics of older adults”– there is a need for consistency. And I still suggest that this sounds like a very general and potentially ageist statement in and of itself. Should it be “perceived characteristics” or “associated characteristics”?
Response 5: I deeply agree with your suggestion. Therefore, I modified “ Difficulties related to the perceived characteristics of older adults” just like the characteristics of older adults as subjectively perceived by the participants, in the abstract as well.
|

Reviewer 2 Report
Comments and Suggestions for Authors
Please, see the attachment.

Author Response
Abstract.
ï¾· I suggested to rewrite a little the first sentence (lines 7-10) for greater clarity, as follows: “This study focused on the experiences and perceptions of ageism by geriatric healthcare professionals. The research aimed to identify related influencing factors, to explore the desired attitudes of geriatric healthcare professionals, and to identify institutional changes required for age-integrated efforts and strategies to eliminate these barriers”. Author states he did the suggested revision but I do not see this in the revised paper.
Response 1: I agree. However, in the results section, I identified the causal and intervening conditions, which were further explored in the discussion to deepen the analysis. In the conclusion, I presented two distinct policy recommendations. While it may still be insufficient, I would appreciate it if you could consider these aspects in understanding the points you mentioned.”
ï¾· I would also suggest to rewrite/simplify this sentence (lines 11-13) as follows: “Comprehensive interviews were conducted in South Korea between August 2023 and November 2023 with 2 physicians and 8 nurses, with each session lasting approximately 1 hour”
Response 2: I agree. I modified,”Interviews with 2 physicians and 8 nurses were conducted in South Korea from August to November 2023, with each session lasting about 1 hour”
Materials and Methods.
ï¾· Firstly, Author should split this paragraph at least in two sub-par., i.e., 2.1. Study Design, Data Collection, and Ethics; and 2.2. Data Analysis.
ï¾· In 2.1. Study Design, Data Collection, and Ethics Author should:
ï¾§ add where data were collected, i.e., in South Korea;
-->Response 3: I agree, I added, “at their institutional offices in South Korea”
ï¾§ clarify the following: Author states that 25 interviews were collected and each participant had 1~2 interviews. If participants are 10 (2 physicians and 8 nurses) and each participant had 1~2 interviews, a maximum of 20 interviews should results. Please, clarify.
--> Response 4: I agree, I modified,”Sampling continued until theoretical saturation was achieved after the first interviews, with additional telephone interviews conducted for participants whose responses were unclear in content or meaning. Each participant’s initial interview lasted approximately one hour and was conducted at their institutional offices in South Korea.”
ï¾§ In 2.2. Data Analysis Author should:
ï¾§ to clarify if a software for qualitative data analysis was used;
--> Response 5: Content analysis was conducted using Excel, and the transcribed data were analyzed line by line.
ï¾§ to add that the analysis of results was integrated by verbatim statements which emerged in the transcription of the interviews;
ï¾§ to specify that each quotation (at least I suppose so) was coded by inserting the type of respondent (nurse or doctor) and the progressive interview numbering by alphabetical letters (2 Doctors from A to B) (8 Nurses from A to H).
---> Response 6: I believe your suggestion helps ensure that the content of the paper is conveyed more rigorously. However, I have already included the identifiers you requested for each quotation. Additionally, based on my review of similar studies, it appears that it is not always necessary to include the respondent identifiers for the derived codes. I kindly ask for your understanding and consideration on this matter.
Results.
ï¾§ By my opinion, it is useful to explain/anticipate in the first paragraph of Result also what is a central phenomenon, and thus move from par. 3.1, lines 174-178, the following sentence: “a central phenomenon represents 'what is going on here' and is a central thought or event controlled by action/interactional strategies (Strauss & Corbin, 1990). In other words, it refers to the central thoughts of healthcare professionals that are formed by causal and contextual conditions”;
--> Response 9: I appreciate your suggestion, but my intention was to use the first paragraph of the results section to provide a brief summary of the overall findings. The detailed explanation of the six distinct phenomena and a more thorough analysis were planned for the individual subsections.
If it is still unclear, please feel free to let me know.
ï¾§ Line 439: I read “older patients elderly” (this seems a typo).
--> Response 8: I modified, “Elderly patients”
Conclusions: I do not understand the first sentence “The placement of paragraphs was revised according to their importance and the point was made clearer as your advie”. It seems the response to a reviewer. Moreover, a full stop seems needed at the end of this sentence “Second, to ensure that the previous proposal is implemented effectively”
--> Response 9: I agree, I mocified it. Second, to ensure that the previous proposal is implemented effectively. T”he strategies
Informed Consent Statement. The sentence “The study was conducted in accordance with the Declaration of Helsinki” should be moved to the section “Institutional Review Board Statement”.
--> Response 10: I moved as you advised.
References should be put in the right format (numbers in square brackets in the text, and in the style requested by the Journal in the final reference list at the end of the paper).
--> Response 11: I changed all.
I sincerely appreciate your thorough review and valuable advice on my paper. Your feedback has significantly contributed to making it a better manuscript

Round 3
Reviewer 1 Report
Comments and Suggestions for Authors
Thank you for responding to my previous comments.
Unfortunately, it is my view that further work is needed on this paper. I am concerned that your description of the participants and the interview process has significantly changed over the review process.
As previoulsy mentioned, throughout, the paper needs to be checked for grammar and readability.
For example, from line 105:
The study used purposeful sampling, with the initial participant recommended by my previous colleagues, then other 10 participants were also introduced to the researcher in a snowball sampling by the previous participants. During a phone call, the researcher introducedhimself explained the aims of the study to them, and arranged an appointment with them if they expressed a verbal willingness to participate in this research.
“then other 10 participants” – should this be “a further 9 participants…”
“introducedhimself” – should this be “introduced herself…”
Line 141 says “Two female researchers conducted the study…” Should the other researcher be mentioned in the acknowledgements? What was their role?
There are grammatical errors on the Conclusion (in particular the first and third sentence).
These are just examples. Please check the whole paper in detail.
Comments on the Quality of English Language
Throughout the paper needs to be checked for grammar and readability.
Author Response
Following your advice, I took the time to carefully and thoroughly review the manuscript. Thank you for your valuable feedback and guidance.
Unfortunately, it is my view that further work is needed on this paper. I am concerned that your description of the participants and the interview process has significantly changed over the review process.
Response 1: I apologize for the confusion regarding the number of research participants. I have reviewed the entire paper and corrected this issue.
The study used purposeful sampling, with the initial participant recommended by my previous colleagues, then other 10 participants were also introduced to the researcher in a snowball sampling by the previous participants. During a phone call, the researcher introducedhimself explained the aims of the study to them, and arranged an appointment with them if they expressed a verbal willingness to participate in this research.
-> “then nine other participants were also introduced to the researcher in a snowball
sampling method” on line 107 “introduced himself,” line 109
Comments 2: As previoulsy mentioned, throughout, the paper needs to be checked for grammar and readability.
Response 2: Therefore, I have improved the readability of the entire paper, corrected typos, and revised the punctuation marks. The revisions have been marked in red and were made without altering the original meaning.
--> I put “coverage for” line 38
Despite not always being detrimental, such beliefs and attitudes can have undesirable consequences when they are understood and applied solely on the basis of age, disregarding unique characteristics among individuals [4]
--> While not always detrimental, such beliefs and attitudes can lead to undesirable consequences when applied solely based on age, disregarding unique characteristics [4].line 50
patients and to explore --> patients and “exploring” ine 91
Revised to improve readability.
“Based on grounded theory, open coding was conducted to closely examine the data, identify phenomena, label them, and categorize them. “
--> “Based on grounded theory, open coding was conducted to closely examine the data, identify, label, and categorize phenomena.” line 120
“and worries about old age create anxiety in younger people[15].”
--> “creating additional anxiety about aging itself [15].” line 355
Comment 3: Line 141 says “Two female researchers conducted the study…” Should the other researcher be mentioned in the acknowledgements? What was their role?
Response 3: I agree, so I added.
Acknowledgments: I would like to express our sincere gratitude to Young Ko and Jandi Kim for their valuable contributions in this study. (line 414)
Comment 4: There are grammatical errors on the Conclusion (in particular the first and third sentence).
Response 4: I agree, so I have revised the third sentence to enhance its natural flow and readability as follows.
“According to the results of this study, the following some suggestions should be made.”
--> “Based on this study's results, the following suggestions are proposed.”
“Second, to ensure that the previous proposal is implemented effectively, strategies to address ageism in healthcare settings should include the establishment of hospital environments, staffing, and training that consider specialized treatment for geriatric patients, with nursing practices that are customized to the needs of older patients.”
-> Second, to ensure that the previous proposal is implemented effectively, strategies to address ageism in healthcare settings should include the establishment of hospital environments, staffing, and training that consider specialized treatment for geriatric patients, with nursing practices that are customized to the needs of older patients.
